# INDUCING COOPERATION VIA LEARNING TO RESHAPE REWARDS IN SEMI-COOPERATIVE MULTI-AGENT REINFORCEMENT LEARNING

## ABSTRACT

We propose a deep reinforcement learning algorithm for semi-cooperative multi-agent tasks, where agents are equipped with their separate reward functions, yet with willingness to cooperate. Under these semi-cooperative scenarios, popular methods of centralized training with decentralized execution for inducing cooperation and removing the non-stationarity problem do not work well due to lack of a common shared reward as well as inscalability in centralized training. Our algorithm, called Peer Evaluation based Dual DQN (PED-DQN), proposes to give *peer evaluation signals* to observed agents, which quantifies how they feel about a certain transition. This exchange of peer evaluation over time turns out to render agents to gradually reshape their reward functions so that their action choices from the myopic best-response tend to result in the good joint action with high cooperation. This evaluation-based method also allows flexible and scalable training by not assuming knowledge of the number of other agents and their observation and action spaces. We provide the performance evaluation of PED-DQN for the scenarios ranging from a simple two-person prisoner's dilemma to more complex semi-cooperative multi-agent tasks. In special cases where agents share a common reward function as in the centralized training methods, we show that inter-agent evaluation leads to better performance.

## 1 INTRODUCTION

Recent advances in reinforcement learning have been focused on the utilization of deep learning techniques. These deep learning methods have been applied to several challenging tasks such as playing games (Mnih et al., 2015; Silver et al., 2016), and robotic control (Gu et al., 2017). However, most of these recent techniques were developed under the assumption of controlling a single learning agent. Applications such as robot swarms (Yogeswaran et al., 2013; Huttenrauch et al., 2017) and network routing (Ye et al., 2015) are better modeled as multi-agent systems, where most RL methods designed for single-agent purposes do not directly translate to their multi-agent counterparts. A prominent reason is that other learning agents are usually regarded as part of the environment, where it is difficult to learn cooperation because a small change in one agent's policy may cause another agent's policy to perform sub-optimally (i.e., non-stationarity).

Many approaches have been proposed in order to induce cooperation among agents using multi-agent reinforcement learning. A popular approach, referred to as centralized training and distributed execution (CTDE), assumes that agents are fully cooperative, possibly controlled by a single administrative domain. It uses a centralized training module with a shared reward function that can be detached from the individual policies during execution. Using a centralized module, they address the non-stationarity problem and instill in agents the context of cooperation, which naturally appears in distributed execution after training. Recent related work includes COMA (Foerster et al., 2018), VDN (Sunehag et al., 2018), QMIX (Rashid et al., 2018), CommNet (Sukhbaatar et al., 2016), BiCNet (Peng et al., 2017), and DIAL (Foerster et al., 2016), just to name a few.

However, CTDE-based multi-agent reinforcement learning (MARL) has a couple of limitations. First, many other multi-agent tasks exist with semi-cooperative agents. A semi-cooperative agent naturally has its own separate reward function (the shared reward scenario is a special case), but is

willing to cooperate if an incentive for cooperation[1] is appropriately provided. Semi-cooperation is often more practical due to the complexity of defining a global reward function, or the agents are controlled under different administrative domains. Second, CTDE-based methods involve the concerns of scalability and sampling inefficiency. The input for the centralized neural network involves the concatenation of the observations and actions, and optionally, the full state. Since the order of the agents in the input vector is likely to be pre-determined, the number of samples needed in order to obtain all the possible action combinations for the same state increases drastically. The concatenation of each agent's information is also proportional to the number of agents, which is often large depending on the task. This problem can be disregarded if a central state whose dimensionality is not defined by the number of agents can be used as a sole input to the central module such as Central-V (Foerster et al., 2018). MADDPG (Lowe et al., 2017) addresses individual reward functions, yet through 'pseudo-centralization' by exchanging each agent's observation and action during training, still suffering from inscalability.

In this paper, we propose a deep MARL that addresses aforementioned two concerns. The key idea is to look at the problem from the mechanism design's perspective in game theory. In semi-cooperative sequential tasks, agents will naturally choose the action that is the best response to its current observation and local reward experience. In our approach, we gradually change the game that agents play so that their local-view actions from the best responses become closer to a socially-good joint action. To this end, each agent receives simple peer evaluations in the form of small feedback messages from nearby agents. These peer evaluations quantify how an agent *feels* about a certain transition, rendering each agent reshape its reward function, so as for agents' local best responses to result in the global cooperation. This approach is empowered to be scalable and flexible, because (i) the knowledge of the number of existing agents beyond an agent's observation is not required, and (ii) this method does not require agents to share states and actions.

We instantiate our idea to a value-based algorithm and propose a Peer-Evaluation based Dual DQN (PED-DQN). We use two DQNs trained on a separate timescale, defined by the length of their experience replays. One DQN, called Action DQN (A-DQN), is trained using the evaluated rewards from the shorter experience replay, while another DQN, called Mission-DQN (M-SDQN) is trained using the agent's base reward from the long-term experience replay. We use the A-DQN to decide the behavior of the agents, while the M-DQN is used for incentivization. We demonstrate the performance of PED-DQN by comparing it with two variants of QMIX and independent DQN on various levels of cooperation of the predator-prey environment.

**Related Work** Multi-agent reinforcement learning (MARL) with decentralized execution has been studied extensively (Busoniu et al., 2008). One of the earlier approaches in fully decentralized deep MARL is Tampuu et al. (2017) that used independent Q-learning in Deep Q-Networks. Other fully decentralized approaches were mainly focused on solving the non-stationarity problem in cooperative agents. This problem is especially prevalent in fully decentralized MARL due to the agents' lack of ability to communicate. Omidshafiei et al. (2017) and Palmer et al. (2017) proposed a dynamic learning rate that depends on the TD-error.

Multiple studies about learning communication protocols have been published (Das et al., 2017; Lazaridou et al., 2016; Mordatch & Abbeel, 2017). Closely related to our line of research is by Foerster et al. (2016) which proposed to push gradients from one agent to another through the communication channel. This can be regarded a form of providing a feedback from one agent to another. A significant downside to their approach is inscalability when all agents are allowed to communicate at the same time. A more scalable communication approach was proposed by CommNet (Sukhbaatar et al., 2016) where agents are allowed to "negotiate" before deciding their actions. This is done by having a message output for each neural network layer and then averaging the messages for each agent which will be fed to the next neural network layer. Peng et al. (2017) suggest using Bi-directional RNNs before selecting the actions. Their approach induces a predetermined flow of information since they used the Bi-directional RNN as the communication channel between agents.

More sophisticated approaches use centralized training with decentralized execution. MADDPG (Lowe et al., 2017) proposed an algorithm, which includes the state-action pairs of the other agents as an input to the critic network. This can be done because the critic network is not used during execution. However, the complexity of training the critic network increases as the number of agent

---

[1]In cooperative game theory, Nash bargaining solution (Nash, 1950) and Shapley value (Shapley, 1953) are the axiomatic rules for fair distribution of cooperation incentive. The way of allocating cooperation incentive to agents is beyond the scope of this paper.

increases. A similar approach is COMA (Foerster et al., 2018), which tries to solve the credit assignment problem. However, COMA assumes a fully-cooperative scenario in which there is a common, shared reward among all the agents. For value-based approaches, QMIX (Rashid et al., 2018) and VDN (Sunehag et al., 2018) uses some centralized extension to DQN for value function factorization. Similar to Foerster et al. (2016) and Mordatch & Abbeel (2017), we allow communication among agents. However, instead of using messages as supplementary observations, we use these messages to communicate how an agent evaluates a certain transition.

Leibo et al. (2017) studied the environment where each agent has a separate reward function. In particular, agents face a social dilemma where mutual cooperation may give them higher rewards, but they have no guarantee of other agent's cooperation. Hughes et al. (2018) attempts to solve these social dilemmas by explicitly reshaping the rewards using inequity aversion terms. Their method induces cooperation from the agent itself, whereas our method reshapes the payoff function of each agent based on peer evaluations.

## 2 BACKGROUND

### 2.1 STOCHASTIC GAME MODEL

We consider a semi-cooperative multi-agent task by modeling it with a stochastic game. We consider $n$ agents, indexed by $a \in \mathcal{A} = \{1, 2, ..., n\}$. We denote $a^{-1}$ as the set of other agents $\mathcal{A} \setminus a$. Since we assume the possibility of heterogeneous agents, each agent $a$ has a set of actions $U_a$. At each time step, each agent has a partial observation $o_a$, which it uses for its policy $\pi_a(u_a|o_a)$ to decide its action $u_a \in U_a$. The set of observations and states from all agents at a time is denoted by $O$ and $S$, respectively. A joint action of all the agents is denoted by $\boldsymbol{u} \in \mathcal{U}$, where $\mathcal{U} := U_1 \otimes \cdots \otimes U_n$ is the set of all joint actions. We represent the true state as $\boldsymbol{s} \in \mathcal{S}$. Once each agent takes its action, forming a joint action, say $\boldsymbol{u}$, a transition to the next state will occur, which follows a transition function $P(\boldsymbol{s}'|\boldsymbol{s}, \boldsymbol{u})$. Each agent will then receive a reward $r_a(\boldsymbol{s}, \boldsymbol{u})$ that may differ across agents. The agents' goal is to maximize its own total expected return $R_a = \mathbb{E}[\sum_{t=0}^{T} \gamma^t r_a^t]$ where $\gamma$ is the discount factor and $T$ is the length of an episode. Throughout this paper, we use time step '$t$' in the superscript and agent index '$a$' in the subscript, in all notations having their dependence.

### 2.2 DEEP Q-NETWORKS

Deep Q-Network (DQN) is a widely accepted deep reinforcement learning method which directly approximates the optimal Q-function $Q_{\pi^\star}(\boldsymbol{s}, \boldsymbol{u})$ via a deep neural network $Q(\boldsymbol{s}, \boldsymbol{u}; \theta)$ with parameter $\theta$. The Q-function of the optimal policy $\pi^\star$ satisfies $Q_{\pi^\star}(\boldsymbol{s}, \boldsymbol{u}) = \mathbb{E}[R(\boldsymbol{s}, \boldsymbol{u}) + \gamma \max_{\boldsymbol{u}' \in \mathcal{U}} Q_{\pi^\star}(\boldsymbol{s}', \boldsymbol{u}')]$ where $\boldsymbol{s}'$ is the next state. One tries to approximate $Q_{\pi^\star}(\boldsymbol{s}, \boldsymbol{u})$ by a deep neural network $Q(\boldsymbol{s}, \boldsymbol{u}; \theta)$ with parameter $\theta$. The network optimizes the loss function defined as $l(\boldsymbol{s}, \boldsymbol{u}, r, \boldsymbol{s}'; \theta) := (Q(\boldsymbol{s}, \boldsymbol{u}; \theta) - (r + \gamma \max_{\boldsymbol{u}' \in \mathcal{U}} Q(\boldsymbol{s}', \boldsymbol{u}'; \theta')))^2$, where $\theta'$ is the target network parameter which is slowly updated with the current network parameter $\theta$. During the training, the agent chooses the best action under its current Q-network parameter $\theta$, i.e. it chooses $\boldsymbol{u} = \arg\max_{\boldsymbol{u}'} Q(\boldsymbol{s}, \boldsymbol{u}'; \theta)$ for state $s$ and updates its parameter $\theta$ iteratively. For the practical stability of the DQN, people add the experience replay which stores the experiences $(\boldsymbol{s}, \boldsymbol{u}, r, \boldsymbol{s}')$ (Mnih et al., 2015).

## 3 METHODS

In this section, we elaborate on our idea of encouraging cooperation between agents without the need for centralization. The mechanism design hails from the concept of peer evaluation, in which agents are able to communicate their assessment of other agents' performance. In the next subsections, we discuss the details on the network architecture of and the training method of our proposed mechanism, for which we start by presenting how peer evaluations are quantified.

### 3.1 PEER EVALUATION BASED REWARD SHAPING

**Temporal Difference as Peer Evaluation** The concept is that at each time step $t$, each agent $a$ gives an evaluation to its observable agents, which we denote by the set $K_a^t$, when other agents $a^{-1}$ do actions that increases/decreases $a$'s reward. Then, using these evaluation exchange, agent $a$ will also receive peer evaluations from the agents in $K_a^t$, which is used to implicitly reshape its

own reward function. The role of $a$'s evaluation is to to quantify how much help or harm the joint actions $\boldsymbol{u} = (u_a, u_{a^{-1}})$ did with respect to $a$'s reward function, for which we use the temporal difference (TD). The TD seems to be a good candidate of peer evaluation, since it is a good measure of error in the estimation of state values. However, this interpretation of the TD is correct only in stationary environments, where $Q(s, u)$ is correct if the policies are stationary. In non-stationary environments like our setting, changes in the policies of the other agents significantly affect the TD as the Q-values are fitted according to the previous transition probabilities, which are in turn defined by the joint policies. Thus, our idea is to attribute the local TD error to the change in other agents' policy. Using this intuition, we define the evaluation given by agent $a$ as:

$$z_a^t(o_a^t, o^{t+1}, u^t) = r_a + \left(\text{NOT\_DONE} \times \gamma \max_u Q_a(o_a^{t+1}, u; \theta_a^t)\right) - Q_a(o_a^t, u_a^t; \theta_a^t), \tag{1}$$

where $\text{NOT\_DONE} = 0$ when the episode is terminated, and 1 otherwise.

The choice of local TD as a peer evaluation signal comes from the need to be able to evaluate the transition without explicit knowledge about the actions done by the observed agents since the number of encountered agents may vary over time. We argue that it is more sample efficient to learn the value of the transition itself through the agent's local observation than to tie the value of the transition to some combination of the local observations and joint actions. This is because partially-observed transitions involving different agents are not considered distinct. Thus, there is no need to get samples of the same interaction for every other agent.

**How to Reshape Rewards** Since agents are basically reward-driven, a candidate approach is to reshape the reward function such that it gives a higher reward when the agents are cooperating. Those reshaped rewards should depend on a given task and other environmental factors. Thus, we learn how to reshape it for cooperation. At each time step $t$, each agent $a$ will now use the reshaped reward $\hat{r}_a^t$ instead of the base reward $r_a$, with the goal of approximating the following optimal Q-function:

$$Q_a^\star(o_a^t, u^t) = \mathbb{E}\left[\hat{r}_a^t + \gamma \max_u Q_a^\star(o_a^{t+1}, u)\right]. \tag{2}$$

The design of the reshaped reward $\hat{r}_a^t$ is of significant importance, since it will determine how coordinated the converged game will be, i.e., each agent local best response leads to the global cooperation. In our design, We use other agents' evaluation of the transition, and define the following *evaluated reward* $\hat{r}_a^t$ as:

$$\hat{r}_a^t = r_a + \text{sgn}(Z_a^t) \times \min(|Z_a^t|, |Z_a^t - z_a^t|, |Z_a^t + z_a^t|), \tag{3}$$

where

$$Z_a^t = \frac{\beta}{|K_a^t|} \sum_{k \in K_a^t} z_k^t. \tag{4}$$

In the above, $z_a^t$ is the agent's given evaluation as in (1), $Z_a^t$ is the agent's received *aggregated peer evaluation*, $K_a^t$ is the set of observed agents that gave agent $a$ an evaluation, and $\beta$ is the weight hyperparameter quantifying how each agent is willing to reshape its reward. Note that the case when $\beta = 0$ corresponds to when agents are totally selfish. Note that $\text{sgn}(Z_a)$ is the sign of $Z_a$.

In (3), one can choose a simpler formulation $\hat{r}_a = r_a + Z_a$, but we apply the following scheme for better stability. The minimization term attempts to take into account the agent's own given evaluation. This is to avoid overestimating the social value when $Z_a$ and $z_a$ go towards the same direction, and to avoid a sudden change of direction when having different signs. In (4), mainly for simplicity we use just an average of the agent's received evaluations, which may reduce the magnitude of the evaluations when the majority of evaluations has the same sign. However, other schemes for aggregating peer evaluations such as individually-weighted evaluations may be applied, left for a future study.

## 3.2 PEER-EVALUATION BASED DUAL DQN (PED-DQN)

**Dual DQN** We now describe the design of our architecture called *Peer Evaluation based Dual DQN* (PED-DQN), which is composed of two timescale-separated DQNs. Each learning agent contains an *Action DQN* (A-DQN) and a *Mission DQN* (M-DQN), which are used for the action-selection and peer evaluation, respectively. The A-DQN considers other agent's feedback when

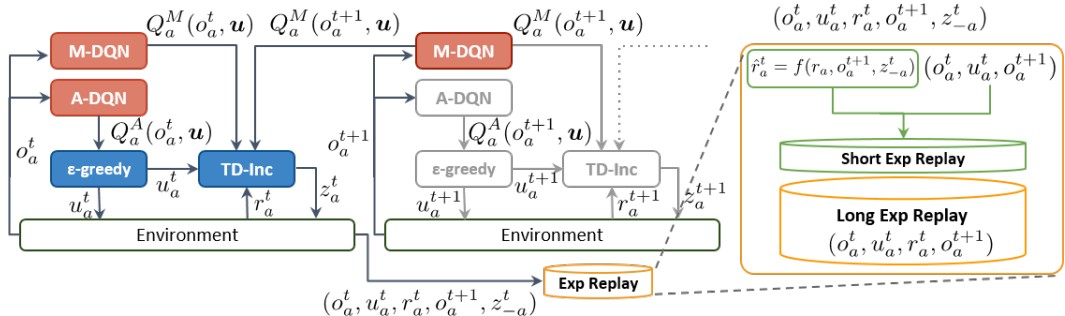

Figure 1: The flow of information during the training. Action DQN (A-DQN) outputs the Q-values of the observation-action pairs at time $t$. We select an action $u_a^t$ using an $\epsilon$-greedy policy and then receive the reward $r_a^t$ and the next observation $o_a^{t+1}$. We then calculate the evaluations by using the TD of Mission DQN (M-DQN). After exchanging evaluations with observed agents, we calculate the evaluated reward and add the transitions to their respective replay buffers.

deciding on an action and is trained on the evaluated reward $\hat{r}_a$. Thus, the A-DQN tries to minimize the loss function:

$$l_a^A(o_a^t, u_a^t, \hat{r}_a^t, o^{t+1}) = \left( Q_a^A(o^t, u^t; \theta_a^t) - (\hat{r}_a^t + \gamma \max_u Q_a^A(o^{t+1}, u; \theta_a'^t)) \right)^2. \qquad (5)$$

In contrast, M-DQN is trained according to its own "mission" defined by the base reward $r_a$, utilizing the classical DQN loss. Using the M-DQN, we calculate the agent's evaluation as defined in (1), making $Q_a^A$ a function of $Q_{-a}^M$. The flow of data in PED-DQN is illustrated in Figure 1.

The problem with using a single DQN is that the peer evaluations are too unstable. This is because the TD is fitted according to the latest policy of the other agents. For the same observations and joint actions, the incentives given at times $t$ and $t + k$ might be too different even for a small $k$. This confuses the receiving agents and may cause the DQNs to fail to converge. In addition, a single DQN will not be able to differentiate between peer evaluations and actual rewards. Therefore, given incentives are heavily affected by the evaluations received in the previous training steps.

An alternate design would be to have two sets of outputs for a single DQN. This may solve the separation of the incentives and the base rewards, but this does not completely solve the instability of the incentives. In order to address it, we have to use a separate DQN with a different timescale. The key idea is that A-DQN is fitted according to the latest joint policies of the agents, while M-DQN learns the Q-values according to the different joint policies of the agents across time. In addition, we only train M-DQN using the base rewards. This allows the agents to give evaluations that are not heavily influenced by the received evaluations.

Although there are no theoretical convergence guarantees for the DQN, the convergence of the algorithm depends on the convergence of the M-DQN. Once the M-DQN converges, the evaluated rewards become more stable, which in turn stabilizes the rewards perceived by the A-DQN.

**Dual Replay Buffer and TD-error** To achieve timescale differentiation between A-DQN and M-DQN, we employ dual replay buffers with different capacities. A short replay buffer is used to store transitions for training A-DQN, while a long replay buffer is used for the M-DQN. If the long replay buffer is long enough, we can approximately say that M-DQN outputs the *expected returns* of the agent with respect to the *average policy* of the observed agents. A negative TD means that observed agents' joint actions lead to lower expected returns, so as for the other agents to be penalized.

**Scalability and Flexibility** Since we use two DQNs for a single agent, the number of parameters to be trained is also doubled. However, we argue that doubling the number of parameters does not necessarily double the complexity of the task. We note that the scalability problems in a neural network generally come from the complexity of the input, which in our case does not increase since we only use local observations. In addition, the local observation assumption is more flexible to the varying number of observed agents. This is in stark contrast to many existing algorithms with a centralized training module whose input size increases as the number of agents $n$ grows.

### 3.3 Example: Prisoner's Dilemma

We illustrate how our peer evaluation based reward reshaping works using an example of two-person Prisoner's Dilemma (PD), as in Table 1a. It is widely known that the PD game has the property that

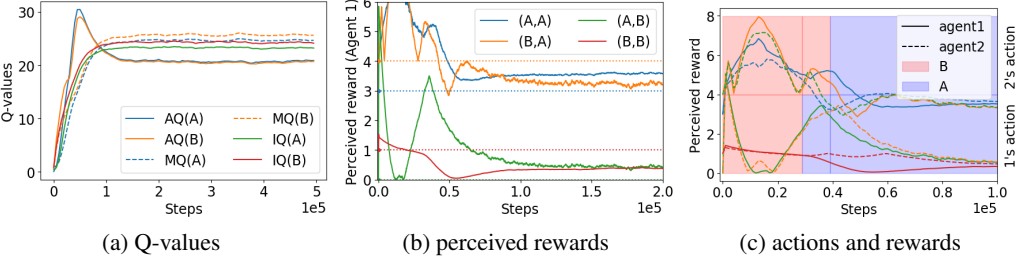

(a) Q-values       (b) perceived rewards       (c) actions and rewards

Figure 2: Evolution of Q-values and perceived rewards for the Prisoner's dilemma

the Nash equilibrium (formed by each agent's best response) differs from the socially optimal joint action. At each step, both agents choose their actions and receive a reward according to their joint actions. The players play the game repeatedly for 500k times. Since it is a stateless game, we only use the tabular Q-learning, where the learning rate $\alpha = 0.001$, the discount factor $\gamma = 0.9$, the evaluation weight $\beta = 1.4$. Agents use an $\epsilon$-greedy policy where we linearly decrease $\epsilon$ from 1.0 to 0.05 in the first 50k steps. For testing purposes, we keep track of the evaluated rewards by simulating all possible combination of actions.

When agents use the best-response strategy, they will both choose action B, which is a sub-optimal point. This is observed when we use independent Q-learning. However, when we use peer evaluation signals, the perceived payoff tables are effectively changed. The Q-values and reward evolution are

|   | A | B |
|---|---|---|
| A | 3, 3 | 0, 4 |
| B | 4, 0 | 1, 1 |

|   | A | B |
|---|---|---|
| A | 3.6, 3.6 | 0.4, 3.2 |
| B | 3.2, 0.4 | 0.4, 0.4 |

(a) original payoff       (b) perceived payoff

Table 1: Prisoner's dilemma

shown in Figures 2a and 2b, respectively. Only agent 1's values are shown since the game is symmetric and the plots are similar. A zoomed-in plot of the evaluated rewards is shown in Figure 2c. Notice that at around $t = 22k$, the game has already changed such that it has a single global optimum. However, both agents still have not learned to choose action $A$. To encourage the other agent into choosing $A$, they keep increasing the incentives for $A$ whenever their actions don't match, and keep increasing the penalty for $B$ in all cases. This goes on until they realize that the other agent has changed policies or the Mission Q-table converges. Once the agents start acting cooperatively, the evaluation decreases to a value that is just high enough to prevent agents from changing their policies as can be seen in Figure 2b, and Table 1b.

## 4 EXPERIMENTS

### 4.1 ENVIRONMENTS

**Fully-cooperative Pursuit** A popular performance evaluation environment in cooperative multi-agent systems is *pursuit* (a.k.a. *predator-prey*). The environment is an $n \times n$ grid world, where each cell is either empty, a wall, or occupied by a predator or a prey. We only trained the predator while the prey is a rule-based agent whose policy is to move to an adjacent empty cell. The goal is to surround the prey with all the predators. A larger number of predators would require more cooperation since they all have to be in the $3 \times 3$ area centered around the prey. Each agent has five possible actions corresponding to `north`, `south`, `east`, `west`, and `stay`. Agents can only observe a $5 \times 5$ grid centered at their own coordinates. Observations at the edge of the map are padded with walls. Predators are rewarded with value 10 when they complete the task.

**Selfish Quota-based Pursuit** To show our method's effectiveness for semi-cooperative agents, we modified the pursuit scenario to allow individual rewards. For each episode, each predator will be assigned a number of preys to capture, indicated by the *quota*. Only those within the $3 \times 3$ area centered at the prey will be credited for the capture. A predator will only get a positive reward when it achieves its quota. In this scenario, the agents get a penalty of $-0.1$, whenever they chose actions aside from `stay`. This means that there is no reason to move after the agent has achieved its own quota. The episode is done when all the agents achieve their respective quotas or the maximum number of steps is reached. A prey will regenerate in a random location if the environment runs out of prey. We change the difficulty of the task by varying the map size and the minimum number of agents around the prey in order to capture.

## 4.2 ABLATIONS, ARCHITECTURE, AND TRAINING

We perform ablation experiments to validate our claims that TD-based peer evaluation leads to more prosocial strategies among agents. First, because of the initial stochasticity of received rewards, one may claim that agents are cooperating in order to either shorten the episode's length or get positive rewards from observing other agents. To invalidate this claim, we compare our method with agents that give random evaluation values. Second, we show that TD-based evaluation is a smarter way of shaping cooperation than just naively sharing the rewards by comparing this to the case where $z_a = r_a$. Finally, we show the need for a separate DQN for peer evaluation by comparing it to a scenario where the evaluations come from the TD error of the Action DQN. Since we removed the second DQN, this peer evaluation scheme shall be called PE-DQN.

For our experiments, we used a single convolution layer followed by three fully-connected layers. The convolution layer has 32 units, while the two hidden layers have 64 units each. Agents do not share parameters for the independent DQN (I-DQN) and PED-DQN. The input to the convolution layer is the $5 \times 5 \times 3$ binary encoding of the agent's observations. The quota and current number of preys captured are appended to the flattened output of the convolution layer before passing through the fully-connected layers.

For comparison in the fully-cooperative pursuit, we implemented QMIX with the same architecture as the I-DQN, but allowed parameter sharing. We appended a onehot-vector encoding of the agent index right after the convolution layer. Similarly, we encode the observation using a convolutional layer before passing through QMIX's hypernetworks. We tested using different learning rates, buffer size, and training intervals. We present the best results of each algorithm in the what follows.

## 4.3 RESULTS AND DISCUSSIONS

To quantify the performance in the fully-cooperative pursuit scenario, we use the number of steps per episode. We compare this against two versions of QMIX: (1) *QMIX-full*, which uses the entire map and the coordinate of each agent as input to the hypernetworks, and (2) *QMIX-partial*, which uses only a concatenation of the observations of each agent as input to the hypernetworks.

Figure 3 shows the performance of each algorithm for a different number of predators. When the number of predators is small, *QMIX-partial* performs the best. However, we note that PED-DQN performs the best in scenarios with a larger number of agents. Not surprisingly, the performance of *QMIX-partial* degrades as the number of agents increases. This can be attributed to the increase in the number of inputs needed to train the centralized part of QMIX. Similarly, the *QMIX-full* might not have been able to factorize the q-values properly due to the number of agents.

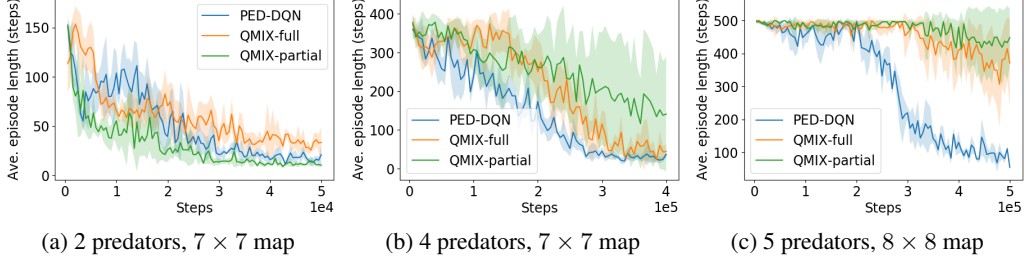

(a) 2 predators, $7 \times 7$ map  (b) 4 predators, $7 \times 7$ map  (c) 5 predators, $8 \times 8$ map

Figure 3: Performance evaluation on fully cooperative pursuit

For the selfish quota-based pursuit, we show the results for four agents and set `quota = 4` for one agent while the rest of the agents only have to capture one prey. We shuffle the quotas for each episode. We compared to two baselines: (1) independent DQN (*I-DQN*), and (2) shared DQN (*S-DQN*), in which the reward of each agent is equal to the summation of the rewards of all the agents. Note that S-DQN practically transforms the game into a fully-cooperative scenario.

We measure the average success rates of the agents in three difficulty levels: (1) pair capture, (2) trio capture, and (3) full capture. In pair and trio captures, the number of predators needed to capture the prey is two and three, respectively. In full capture, the prey must not be able to move in any direction. As shown in Figure 4, PED-DQN achieves a high success rate faster than both baselines. We also note that for pair and trio captures, PED-DQN converges to the same success rate as S-DQN. This means that the peer evaluation has successfully transformed the perceived individual reward function of the agents into a cooperative reward function.

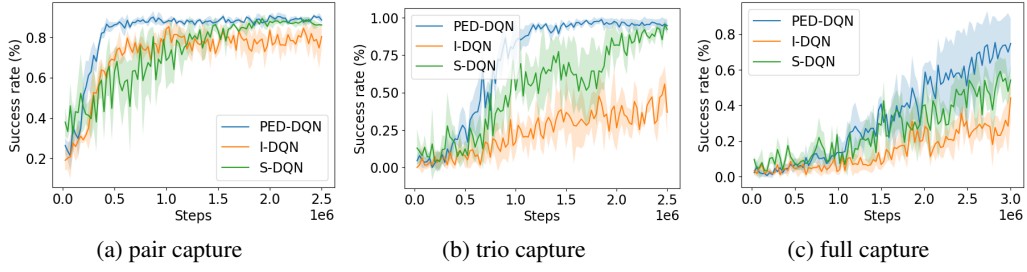

Figure 4: Performance evaluation on quota-based pursuit.

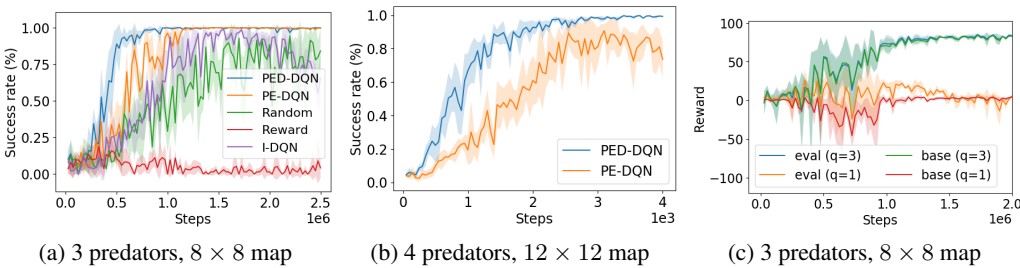

(a) 3 predators, $8 \times 8$ map    (b) 4 predators, $12 \times 12$ map    (c) 3 predators, $8 \times 8$ map

Figure 5: Trio capture ablations and rewards

We performed our ablation experiments on the trio capture scenario. As seen in Figure 5a, the TD-error based peer evaluation provides a stable way of transforming the original game into a cooperative one. Although the random peer evaluation values also improve through time, we expect that the DQNs will not be able to stabilize. We also note that naively sharing rewards among observable agents did not perform well because the agents learned to avoid each other because of the possibility of sharing the -0.1 reward when observing a moving agent. PE-DQN achieves slower convergence, but it works fairly well for some tasks. To show the need for a separate Mission DQN, we employed a more challenging scenario by increasing the map size and the number of agents. As shown in Figure 5b, the PE-DQN's success rate starts to decrease after some time. We conjecture that this is because of the Action DQN's dependence on other agent's evaluation since a small change can cause an alternating positive and negative evaluations among agents.

We now attempt to analyze the evaluated rewards of the agents through training. We ran an experiment on the trio capture scenario with three predators, and fixed their respective quotas as $(3, 1, 1)$. Figure 5c shows the graph of the total base rewards and evaluated rewards of the following two types of agents. The high-quota (HQ) agent receives a reward of 90 when it completes the quota. As seen in Figure 5c, the evaluated rewards of the high-quota agent is almost the same as the base reward. This is because the low-quota (LQ) agents do not need to give a non-zero evaluation to the HQ agent. In addition, once the LQ agents complete their quotas, there is no reason to give a high magnitude evaluation. On the other hand, notice that the evaluated reward of the LQ agents converge to a value that is close to the base reward. This happens because the TD of M-DQN decreases as the long experience replay is filled with episodes where the joint policies of the other agents is to cooperate. Another point of interest is that when the HQ agent gets a lower base reward, the LQ agents also get a lower evaluation, thus confirming the effectiveness of our evaluation scheme.

## 5 CONCLUSION

In this paper, we presented Peer-evaluation based Dual DQN (PED-DQN) in order to induce cooperation among semi-cooperative agents. Through PED-DQN, agents exchange peer evaluation signals, which gradually changes the game so that an agent's myopic best-response leads to a higher global reward. Our results show that using the TD as an inter-agent evaluation smartly reshapes the individual rewards into a global reward that encourages cooperative behaviors among agents. We have also shown that PED-DQN can also work for fully-cooperative scenarios and achieve faster training time. Future works will include smarter aggregation of peer evaluation, as well as run-time peer evaluations in which agents could change policies during execution.

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

SUPPLEMENTARY MATERIAL

## A  EXPERIMENT DETAILS

### A.1  FULLY-COOPERATIVE PURSUIT

The observation of each agent is a $5 \times 5 \times 3$ binary encoding of the $5 \times 5$ grid centered around the agent. The first channel encodes the presence of a wall, the second channel encodes the presence of predators, and the third channel encodes the presence of a prey.

All agents have different neural networks in the Peer-evaluation based Dual DQN (PED-DQN). Each DQN is composed of one convolution layer with 32 units, followed by two hidden layers with 64 units, and finally an output layer corresponding to the actions.

For the QMIX implementation, we used the same network architecture as I-DQN but appended a onehot encoding of the agent's ID to the flattened output of the convolution layer in order to allow parameter sharing. In the QMIX-full version, we used the full encoding of the whole map as an input to a 64-unit convolution layer, and then appended the normalized coordinates of all the predators to the flattened output. We then use that as the input to the hypernetworks. In the QMIX-partial version, we use the individual observations of each agent as an input to a 32-unit convolution layer. We then concatenated the flattened outputs of the observations and used this as the input to the hypernetworks. The mixing network uses two hidden layers with 64 units each. The hypernetworks are constructed according to the required parameters of the mixing networks.

All networks are optimized using Adam. For PED-DQN, I-DQN, and QMIX-partial, we used a learning rate of 0.0001 in all scenarios, except for the 5-predator case where 0.00005 performed best. For the QMIX-full, a learning rate of 0.00005 performed best for all scenarios. We used a discount factor $\gamma = 0.99$. The PED-DQN uses a peer evaluation weight of $\beta = 1.0$. The long experience replay, which is also used by QMIX, has a capacity of 100,000 steps, while the short experience replay contains the recent 1000 steps. Target networks are updated every 50 steps. Episode length and training steps are specified in Table 2.

| no. of agents | map size | training steps | maximum episode length |
|---------------|----------|----------------|------------------------|
| 2 | $7 \times 7$ | 50000 | 200 |
| 4 | $7 \times 7$ | 400000 | 400 |
| 5 | $8 \times 8$ | 500000 | 500 |

Table 2: Training steps and episode lengths

### A.2  SELFISH QUOTA-BASED PURSUIT

The observation of each agent is a $5 \times 5 \times 3$ binary encoding of the $5 \times 5$ grid centered around the agent. The first channel encodes the presence of a wall, the second channel encodes the presence of predators, and third channel encodes the presence of a prey. In addition, the agent keeps track of its quota during the episode and the number of prey it has already captured.

Multiple prey exists in the map. A prey disappears from the map after being captured. A new prey will appear in a random location when all preys disappear before all the agents have achieved their quota.

Agents have different neural networks each. The individual DQNs are composed of one convolution layer with 32 units, followed by two hidden layers with 64 units, and finally an output layer corresponding to the actions. The quota and progress tracker are appended to the flattened output of the convolution layer before it enters the first hidden layer.

All networks are optimized using Adam with a learning rate of 0.0001. We used a discount factor $\gamma = 0.99$ and a maximum episode length of 1000. The PED-DQN uses a peer evaluation weight of $\beta = 1.0$. Since the non-stationarity problem is more apparent in this scenario, agents are trained every 32 steps with only the recent 32 steps as training data. The long experience replay contains the recent 100,000 steps. Target networks are updated every after 50 updates of the current network.

# B ADDITIONAL RESULTS

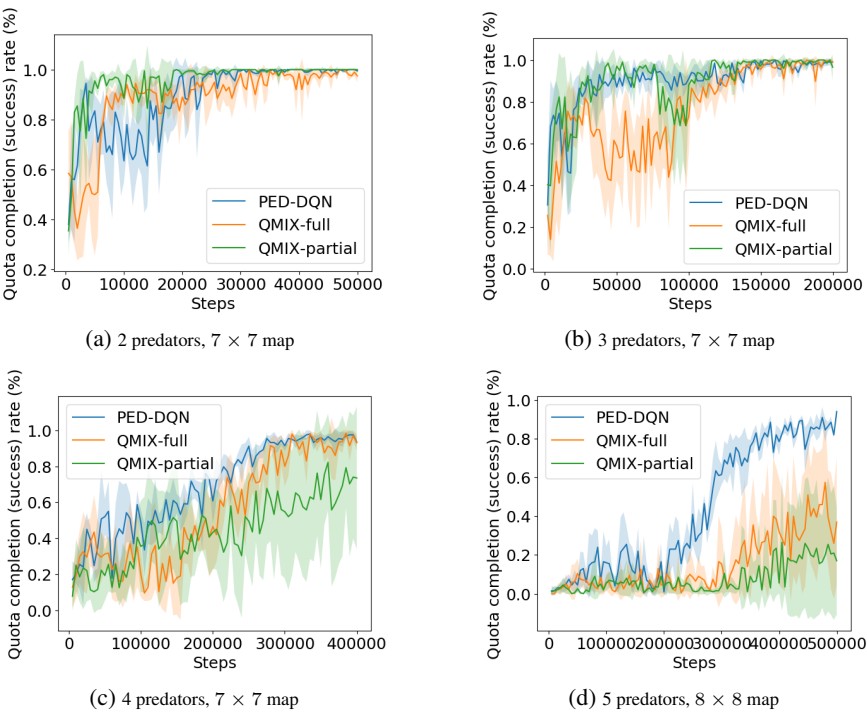

Figure 6: Success rate on fully cooperative pursuit

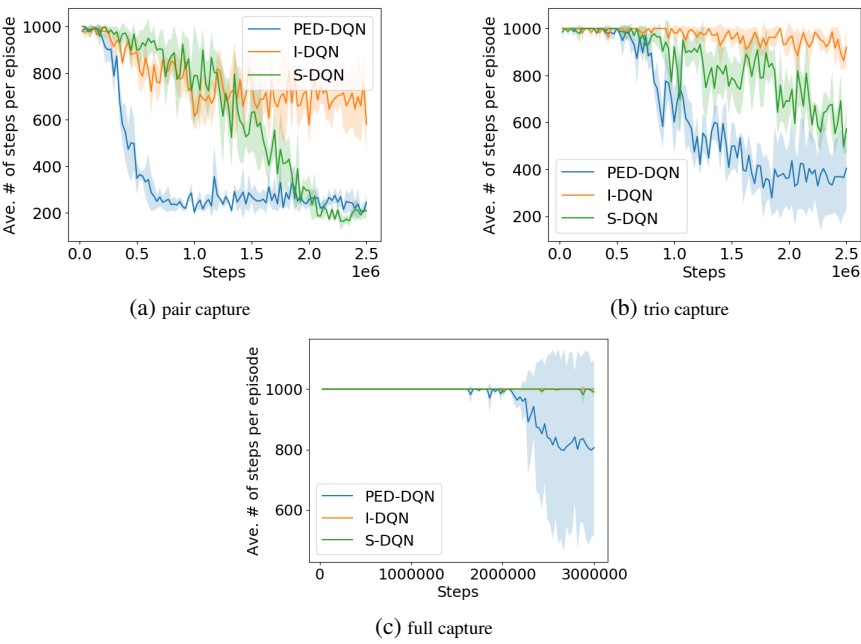

Figure 7: Episode lengths on selfish quota-based pursuit

