# OpenReview forum: "Inducing Cooperation via Learning to reshape rewards in semi-cooperative multi-agent reinforcement learning"
_ICLR.cc/2019/Conference_

### Official Review · AnonReviewer1 · 2018-11-01
**Potentially an intriguing paper, however some key choices are poorly motivated and a few results miss leading.**

**Rating:** 5
**Confidence:** 4

**Review:**

The authors suggest a reward shaping algorithm for multi-agent settings that adds a shaping term based on the TD-error of other agents to the reward. In order to implement this, each agent needs to keep tack of two different value estimates through different DQN networks, one for the unshaped reward and one for the shaped reward.

Points of improvement and questions:
-Can you please motivate the form of the reward shaping suggested in (2) and (3)? It looks very similar to simply taking \hat{r}_a = r_a + sum_{a' not a} z_{'a}. Did you compare against this simple formulation? I think this will basically reduce the method to Value Decomposition Networks (Sunehag ‎2017)
-The results on the prisoners dilemma seem miss-leading: The "peer review" signal effectively changes the game from being self-interested to optimising a joint reward. It's not at all surprising that agents get higher rewards in a single shot dilemma when optimising the joint reward. The same holds for the "Selfish Quota-based Pursuit" - changing the reward function clearly will change the outcome here. Eg. there is a trivial adjustment that adds all other agents rewards to the reward for agent i that will will also resolve any social dilemma.
-What's the point of playing an iterated prisoners dilemma when the last action can't be observed? That seems like a confounding factor. Also, using gamma of 0.9 means the agents' horizon is effectively limited to around 10 steps, making 50k games even more unnecessary.
-"The input for the centralized neural network involves the concatenation of the observations and actions, and optionally, the full state": This is not true. For example, the Central-V baseline in COMA can be implemented by feeding the central state along (without any actions or local observations) into the value-function. It is thus scalable to large numbers of agents.
-The model seems to use a feed-forward policy in a partially observable multi-agent setting. Can you please provide a justification for this choice? Some of the baseline methods you compare against, eg. QMIX, were developed and tested on recurrent policies. Furthermore, independent Q-learning is known to be less stable when using feedfoward networks due to the non-stationarity issues arising (see eg. "Stabilising Experience Replay", ICML 2017, Foerster et al). In it's current form the concerns mentioned outweigh the contributions of the paper.

---

> ### Author Response · Authors · 2018-11-20
> **Reward Shaping, Prisoner's Dilemma, and other details**
>
> Reward shaping in (2) and (3)
>
> The initial design of the reward shaping is similar to your suggested simple formulation \hat{r}_a = r_a + Z_a. The reason we used a subset K_a in (3) is so that the “change” in the reward can be associated with the agents. We think that receiving peer evaluation from unobservable agents is just noise, although this may not be the case for agents with different observation ranges.
>
> The minimization term in (2) takes tries not to overestimate the value of the transition. For example, if agent a thinks that the transition is ‘good’ (z_a > 0) and then it is also incentivized by other agents (Z_a > 0), then agent a only needs a little push because the value update is already going to that direction. On the other hand, if Z_a < 0 and z_a > 0, then it tries to prevent a sudden change in the direction of the update. Of course the change of direction in extreme cases (e.g. Z_a << z_a, Z_a >> z_a) may be inevitable, and we may need to clip the rewards. Without the minimization term, the algorithm still works but the fluctuation of value functions are more apparent.
>
> As of the comment about VDN, in a way, our method looks like the reverse process of VDN but starting with individual rewards instead of a global one. Thus, the values are already decomposed. However, we don’t have any proof that this reduces to VDN.
>
> Prisoner’s Dilemma
>
> Although the last action cannot be observed, the base rewards can imply the action of the opponent. Also, we designed the PD so that the agents do not condition their Q-values on more than 1 state, and thus easier to analyze.
>
> The goal is to reshape the rewards so that the perceived reward of the agents become as cooperative as they can. The willingness to cooperate, \beta, governs this. In the prisoner’s dilemma, we had to use a beta of 1.4 so that they could read the global optimum (3,3).
>
>
> Central-V, QMIX, I-DQN
>
> Thank you for pointing out the error in centralized neural networks. We have updated the introduction appropriately. As for using feed-forward networks, we did not focus on the neural network structure. Similarly, QMIX’s main contribution is the value function factorization. Of course, RNN can be used as an alternative when necessary. For the experiments that involved I-DQN we only used recent buffer [1] for all the algorithms. Of crouse, the M-DQN in our framework used a replay buffer.
>
> [1]Joel Z Leibo, Vinicius Zambaldi, Marc Lanctot, Janusz Marecki, and Thore Graepel.  Multi-agent reinforcement learning in sequential social dilemmas. InProceedings of International Conference on Autonomous Agents and MultiAgent Systems, pp. 464–473, 2017.

---

### Official Review · AnonReviewer2 · 2018-11-01
**Interesting connections to study of social dilemma and role of peer evaluation; experiments not enough to make the scalability claim**

**Rating:** 5
**Confidence:** 4

**Review:**

The paper introduces a DQN based, hierarchical, peer-evaluation scheme for reward design that induces cooperation in semi-cooperative multi-agent RL systems. The key feature of this approach is its scalability since only local “communication” is required -- the number of agents is impertinent; no states and actions are shared between the agents. Moreover this “communication” is bound to be low dimensional since only scalar values are shared and has interesting connections to sociology. Interesting metaphor of “feel” about a transition.

Regarding sgn(Z_a) in Eq2, often DQN based approaches clip their rewards to be between say -1 and 1. The paper says this helps reduce magnitude, but is it just an optimization artifact, or it’s necessary for the reward shaping to work, is slightly unclear.

I agree with the paper’s claim that it’s important for an agent to learn from it’s local observation than to depend on joint actions. However, the sentence “This is because similar partially-observed transitions involving different subsets of agents will require different samples when we assume that agents share some state or action information.” is unclear to me. Is the paper trying to just say that it’s more efficient because what we care about is the value of the transition and different joint actions might have the same transition value because the same change in state occured. However, it seems that paper is making an implicit assumption about how rewards look like. If the rewards are a function of both states and actions, r(s,a) ignoring actions might lead to incorrect approximations.

In Sec 3.2, under scalability and flexibility, I agree with the paper that neural networks are weird and increasing the number of parameters doesn’t necessarily make the task more complex. However the last sentence ignores parameter sharing approaches as in [1], whose input size doesn’t necessarily increase as the number of agents grows. I understand that the authors want to claim that the introduced approach works in non homogeneous settings as well.

I get the point being made, but Table 1 is unclear to me. In my understanding of the notations, Q_a should refer to Action Q-table. But the top row seems to be showing the perceived reward matrix. How does it relate to Mission Q-table and Action Q-table is not obviously clear.

Given all the setup and focus on flexibility and scalability, as I reach the experiment section, I am expecting some bigger experiments compared to a lot of recent MARL papers which often don’t have more two agents. From that perspective the experiments are a bit disappointing. Even if the focus is on pedagogy and therefore pursuit-evasion domain, not only are the maps quite small, the number of agents is not that large (maximum being 5). So it’s hard to confirm whether the scalability claim necessarily make sense here. I would also prefer to see some discussion/intuitions for why the random peer evaluation works as well as it did in Fig 4(a). It doesn’t seem like the problem is that of \beta being too small. But then how is random evaluation able to do so much better than zero evaluation?

Overall it’s definitely an interesting paper. However it needs more experiments to confirm some of its claims about scalability and flexibility.

Minor points
I think the section on application to actor critic is unnecessary and without experiments, hard to say it would actually work that well, given there’s a policy to be learned and the value function being learned is more about variance reduction than actual actions.
In Supplementary, Table 2: map size says 8x7. Which one is correct?

[1]: https://link.springer.com/chapter/10.1007/978-3-319-71682-4_5

---

> ### Author Response · Authors · 2018-11-20
> **Equation (2)'s effect and reward assumptions**
>
> Reward shaping in (2)
>
> The sgn(Z_a) in (2) is there to keep the original sign of the aggregate peer evaluation Z_a which may have changed in the minimization term due to the absolute value operation. We can also clip the peer evaluation values, but it may be difficult to select the correct clipping parameters. The term that actually reduces the magnitude is the minimization term.
>
> The minimization term in (2) takes tries not to overestimate the value of the transition. For example, if agent a thinks that the transition is ‘good’ (z_a > 0) and then it is also incentivized by other agents (Z_a > 0), then agent a only needs a little push because the value update is already going to that direction. On the other hand, if Z_a < 0 and z_a > 0, then it tries to prevent a sudden change in the direction of the update. Of course the change of direction in extreme cases (e.g. Z_a << z_a, Z_a >> z_a) may be inevitable, and we may need to clip the rewards. Without the minimization term, the algorithm still works but the fluctuation of value functions are more apparent.
>
> Reward assumption and observation claim
>
> The statement was under the assumption that we are using a neural network and that each agent has a fixed input index in each other’s NN for their shared message (e.g. action, state, signal). In this case, if agent 1 observed state o_t, and the same observation o_{t+k}, but with different agents involved, the two experiences are treated as different. This can be observed in MADDPG’s critic networks. We have modified the TD as Peer evaluation part to make this clear..
>
> You are correct. Having knowledge of the actions of the other agents may be better. However, even if the reward is a function of both states and actions, the agents can still give a good peer evaluation since the evaluation is a function of the reward. A lower/higher than expected base reward will result in an agent giving a penalty/incentive even if they don’t have explicit knowledge of the actions.
>
> Other Things
>
> Thank you for pointing out the parameter sharing approaches. And yes, we are also claiming that this approach works on heterogeneous agents.
>
> As for the table 1, we added a more comprehensive illustration and discussion of the evolution of the mission-q and action-q.
>
> We also updated figure 4(a) to include independent DQN (zero evaluation). In contrast to what you said, random does not perform so much better than zero evaluation. If you were asking why random is much better than sharing the reward to observable agents (‘reward’ in the plot), it is because agents learned to avoid other agents. Since agents are given a reward of -0.1 for relocating, the observers also get -0.1. We added this in the discussion.

---

### Official Review · AnonReviewer3 · 2018-11-02
**This paper addresses this challenge by introducing a reward-shaping mechanism and incorporating a second DQN technique which is responsible for evaluating other agents performance. No discussion about convergence.**

**Rating:** 5
**Confidence:** 3

**Review:**

This work is well-written, but the quality of some sections can be improved significantly as suggested in the comments. I have a few main concerns that I explain in detailed comments. Among those, the paper argues that the algorithms converge without discussing why. Also, the amount of overestimation of the Q-values are one of my big concerns and not intuitive for me in the game of Prisoner's Dilemma that needs to be justified. For these reasons, I am voting for a weak reject now and conditional on the authors' rebuttal, I might increase my score later.

1) I have a series of questions about Prisoner's Dilemma example. I am curious to see what are the Q-values for t=100000 in PD. Is table 1h shows the converged values? What I am expecting to see is that the Q-values should converge to some values slightly larger than 3, but the values are  ~ 30. It is important to quantify how much bias you add to the optimal solution by reward shaping, and why this difference in the Q-values are observed.

2) One thing that is totally missing is the discussion of convergence of the proposed method. In section 3.4, you say that the Q-values converge, but it is not discussed why we expect convergence. The only place in the paper which I can make a conjecture about the convergence is in figure 4c which implicitly implies the convergence of the Mission DQN, but for the other one, I don't see such an observation. Is it possible to formalize the proposed method in the tabular case and discuss whether the Q-values should converge or not? Also, I would like to see the comparison of the Q-values plots in the experiments for both networks.

3) The intuition behind (2) should be better clarified. An example will be informative. I don't understand what |Z_a| is doing in this formula.

4) One of the main contributions of the paper is proposing the reward shaping mechanism. When I read section 3.3, I was expecting to see some result for policy gradient algorithms as well, but this paper does not analyze these algorithms. That would be very nice to see its performance in PG algorithms though. In such case that you are not going to implement these algorithms, I would suggest moving this section to the end of the paper and add it to a section named discussion and conclusion.

5) Is it possible to change the order of parts where you define $\hat{r}$ with the next part where you define $z_a$? I think that the clarity of this section should be improved. This is just a suggestion to explore. I was confused at the first reading when I saw $z_a$, "evaluation of transition" and then (2) without knowing how you define evaluation and why.

6) Is there any reason that ablation testing is only done for trio case? or you choose it randomly. Does the same behavior hold for other cases too?

7) Why in figure 4a, random is always around zero?

8) What will happen if you pass the location of the agent in addition to its observation? In this way, it is possible to have one  Dual-Q-network shared for all agents. This experiment might be added to the baselines in future revisions.

Minor:
* afore-mentioned -> aforementioned
section 4.2: I-DQN is used before definition
* Is it R_a in (4)?
* I assume that the table 1f-1h are not for the case of using independent Q-learning. Introducing these tables for the first time right after saying "This is observed when we use independent Q-learning" means that these values are coming from independent Q-learning, while they are not as far as I understand. Please make sure that this is correct.
section 4.1: * who's -> whose
* This work is also trying to answer a similar question to yours and should be referenced: "Learning Policy Representations in Multiagent Systems, by Grover et al. 2018"
* Visual illustrations of the game would be helpful in understanding the details of the experiment. Preparing a video of the learned policies also would informative.
-----------------------------------------------
After rebuttal: after reading the answers, I got answers to most of my questions. Some parts of the paper are vague that I see that other reviewers had the same questions. Given the amount of change required to address these modifications, I am not sure about the quality of the final work, so I keep my score the same.

---

> ### Author Response · Authors · 2018-11-20
> **Convergence, ablation tests, and sharing the neural network**
>
> Prisoner’s Dilemma
>
> We made some modifications to the experiments to see the convergence. We extended the steps to 500k but the epsilon goes down from 1.0 to 0.05 only in the first 100k steps.
>
> Since this is a repeated game, the update rule of Q-table is Q[u] = Q[u] + 0.9*maxQ[u]. Given the rewards, it should naturally converge to values around 30. We replaced the tables with a better illustration for the evolution of the Q-values.
>
> Convergence of the method
>
> Since the Mission-DQN is just a regular DQN, the convergence property of that part is the same, which has no theoretical guarantees. If M-DQN converges, then the TD will be consistent with some granularity. When that happens, the hat{r} will also be consistent. We can then treat the A-DQN as a regular DQN with the same convergence property. The experience replay of the M-DQN has to be large enough to accommodate different policies of the other agents. Otherwise, the M-DQN will not be stable (because it is trying to converge to the recent policies) and thus less likely to converge.
>
> Intuition for (2)
>
> Due to the page limit, we could not fit in an example. First, |Z_a| is there for instances where the magnitude of the received evaluation is smaller than the given evaluation. For example, when an agent receives no evaluation at all, then \hat{r} should just be the base reward. On the other hand, if the agent a and other observable agents a’ disagree on the value of transition (i.e. sgn(Z_a) != sgn(z_a)), the agent’s adjustment would be less drastic. However, if they agree on the value of transition (i.e. sgn(Z_a) == sgn(z_a)), \hat{r} could be too large giving the action-Q a large update, only to be decreased again on the next time the transition is observed. This is because mission-Q is also learning. In many cases, the TD error is lower the next time they observe the transition. The formulation \hat{r}_a = r_a + Z_a will also work.
>
> Ablation Testing
>
> Yes, the trio capture was selected randomly. We expect to observe the same trend for the other cases.
>
> Regarding Figure 4(a), we believe you are referring to reward, instead of random. In reward, we give the agents reward as a peer evaluation. This means that every time agent a relocates, other agents that can observe will also receive -0.1. The agents learned to avoid other agents, to avoid sharing this additional -0.1 from the other agents.
>
> Sharing the neural network
> One way of sharing the neural network is to concatenate the agent ID to the observation. However, when we experimented on this, the NN learned to ignore the agent IDs. Although the location information is a different information, this may still lead to the agents having the same policy.

---

### Author Response · Authors · 2018-11-20
**Revisions**

Dear reviewers,

Thank you for the valuable insights. Significant updates have been made in the methods section to make our framework clearer. We have also added some illustrations in the Prisoner's dilemma game so as to see the convergence of the Q-tables. Individual responses have been commented on your own posts.

---

### Meta-Review · Area_Chair1 · 2018-12-13
**Intriguing work, not yet ready for publication.**

**Confidence:** 4
**Recommendation:** Reject

**Metareview:**

This work introduces a reward-shaping scheme for multi-agent settings based on the TD-error of other agents.

Overall, reviewers were positive about the direction and the presentation but had a variety of concerns and questions and felt more experiments were necessary to validate the claims of flexibility and scalability, with results more comparable to the scale of the contemporary multi-agent literature. One note in particular: a feed-forward Q network is used in a partially observable environment, which the authors seemed to dismiss in their rebuttal. I agree with the reviewer that this is an important consideration when comparing to baselines which were developed with recurrent networks in mind.

A revised manuscript addressed concerns with the presentation but did not introduce new results or plots, and reviewers were not convinced to alter their evaluation. There is agreement that this is an interesting paper, so I recommend that the authors conduct a more thorough empirical evaluation and submit to another venue.